# A Water Quality Prediction Model Based on Multi-Task Deep Learning: A Case Study of the Yellow River, China

**Xijuan Wu** [1], **Qiang Zhang** [1], **Fei Wen** [2] **and Ying Qi** [1,*]

1 Department of Computer Science and Engineering, Northwest Normal University, Lanzhou 730070, China
2 Gansu Academy of Eco-Environmental Sciences, Lanzhou 730070, China
* Correspondence: qiying@nwnu.edu.cn

**Abstract:** Water quality prediction is a fundamental and necessary task for the prevention and management of water environment pollution. Due to the fluidity of water, different sections of the same river have similar trends in their water quality. The present water quality prediction methods cannot exploit the correlation between the water quality of each section to deeply capture information because they do not take into account how similar the water quality is between sections. In order to address this issue, this paper constructs a water quality prediction model based on multi-task deep learning, taking the chemical oxygen demand (COD) of the water environment of the Lanzhou portion of the Yellow River as the research object. The multiple sections of correlation are trained and learned in this model at the same time, and the water quality information of each section is shared while retaining their respective heterogeneity, and the hybrid model CNN-LSTM is used for better mining from local to full time series features of water quality information. In comparison to the current single-section water quality prediction, experiments have shown that the model's mean absolute error (MSE) and root mean square error (RMSE) of the predicted value of the model are decreased by 13.2% and 15.5%, respectively, and that it performs better in terms of time stability and generalization.

**Keywords:** water prediction; deep learning; multi-task learning; CNN-LSTM model; chemical oxygen demand

## 1. Introduction

Water quality prediction is the basic content of water quality protection and is an important basis for solving the water resource crisis. In the prevention and control of water pollution [1], the accurate expression of water quality can reflect the pollution status of the water body and the future trend of water quality, to provide a scientific foundation for a specific area to protect water resources from pollution [2]. With the development of a new generation of information technology, more and more scholars are employing intelligent algorithms to build water quality prediction models for water resource pollution prevention and control [3].

For the prediction of water quality, the current research is roughly divided into three types: traditional statistical method prediction, artificial intelligence method prediction and hybrid model method prediction. Traditional statistical methods for forecasting include the Markov model, regression analysis and so on. For example, dissolved oxygen levels in a riverine environment in Calgary, Canada were predicted using a fuzzy linear regression method. This method improves the prediction of low Do and thus can be used for risk analysis of water resources management [4]. Kumar et al. discuss a "Fuzzy River Health Index" (FRHI) that might take into account subjectivities and uncertainties while assessing the health of the Chambal River in India. This study shows that the FRHI produces generally respectable outcomes [5]. Tamm et al. demonstrate the value of linking the rating-curve method and baseflow separation to evaluate the allochthonous dissolved organic carbon

(DOC) load to Lake Võrtsjärv [6]. Gupta et al. discovered that return period and flood frequency analyses with longer period observed streamflow and rainfall data could be used to provide long-term flood forecasting related to extreme rainfall occurrences for the river basin [7]. The impacts of both observed and anticipated changes in the climate on the loading of dissolved inorganic nitrogen (DIN) to lakes and rivers in five European catchments were quantified by Moore et al. using the Generalized Watershed Loading Functions model (GWLF) [8]. However, these traditional predictive models are too shallow to be compatible with highly complex data.

In order to solve this problem, artificial intelligence methods, such as neural network and support vector machine, are gradually applied to the prediction of water quality. Ju and Wang established a least squares support vector machine model to predict the ammonia nitrogen content in water, and the results showed that the effect was better than the neural network model [9]. Sen's slope tests and the Modified Mann–Kendall (MMK) test were used by Pandey et al. to examine the trend of several meteorological variables in the central Indian region's Betwa river basin. In order to manage water resources effectively, it will be useful to know how long-term changes in climatic factors and their spatial temporal distribution pattern have changed [10]. Bui et al. used a machine learning method to forecast water quality and improved the prediction ability compared with the original algorithm [11]. Due to the limited modeling data of dissolved oxygen content in water, in order to solve the complex environment with multiple pollution sources, a support vector machine model was established to forecast the dissolved oxygen concentration in the suburban section of the Wenruitang River [12]. Wu Jing et al. found that Autoregressive Integrated Moving Average (ARIMA) and Genetic Algorithm Wavelet Neural Network (GAWNN) can be well compatible with complex data, reflecting the linearity features of water quality series [13].

However, due to the shortcomings of the neural network, such as poor generalization ability and overfitting, it is unable to handle the variability and complexity of water quality data. As a result, a new technique known as the hybrid model has been developed to optimize the neural network algorithm for forecasting water quality. Jin et al. created a data-driven hybrid model for surface water quality prediction, and the results revealed that compared with the simple model, the prediction accuracy and reliability were improved [14]. Long Short-Term Memory Network (LSTM) has been effectively applied to water quality prediction, and a water quality prediction model based on LSTM has been established, which overcomes the difficult problems of traditional neural networks and obtains better results than traditional algorithms [15]. However, the prediction performance of a single LSTM network model is restricted. In order to enhance the performance of the LSTM water quality prediction model, relevant scholars have explored the hybrid model of LSTM. Due to the complexity of water quality data, Yang et al. established the CNN-LSTM water quality prediction model based on an attention mechanism. The hybrid model CNN-LSTM has a strong ability to solve nonlinear time series prediction problems, and the attention mechanism can capture longer temporal dependencies, and the findings demonstrate that this model outperforms the CNN model alone or LSTM model [16].

The prediction models established in the above discussion and research have been able to predict water quality more accurately. However, the drawback of these prediction methods is that they only predict the water quality indicators of a single section. Due to the fluidity of water, the historical water quality monitoring data show long-term dependency in time series characteristics as well as similarity of dependence between sections [17,18]. Most of the current prediction models ignore the potential dependency similarity between sections and cannot use the correlation between sections to more accurately capture the water quality information between sections, so as to improve the generalization ability, accuracy, convergence speed and applicability, and other aspects have more or less limitations. Therefore, in this context, it is crucial to look at new water quality prediction models and methods. In this case, multi-task learning (MTL) [19,20] is more appropriate. The purpose of MTL is to learn several related prediction tasks simultaneously and communicate the

feature information from various tasks. Compared with single-task learning, MTL can use the relevant information between each task to complement the learned information through mutual sharing between tasks. This learning method can better improve the model effect.

The Yellow River has a significant impact on China's economic growth, social stability and ecological environment protection [21]. However, with the development of economy and the expansion of urban scale, the water pollution has become more and more serious [22]. Therefore, the monitoring and protection of the water quality of the Yellow River is imminent.

Due to the fluidity of the water in the Yellow River, the water quality monitoring sections are correlated, allowing for the simultaneous performance of several prediction tasks. In addition, due to the complexity of water pollution and the nonlinear characteristics of pollutant concentration changes, the accuracy and generalization of water quality prediction have certain shortcomings. Deep learning technology can solve this problem well, and these features can be better obtained through autonomous deep neural network training.

Therefore, this study combines deep learning and multi-task learning [23], taking the Lanzhou portion of the Yellow River as the research object, and proposes a water quality prediction model based on multi-task deep learning (MTL-CNN-LSTM). Methodologically, the research makes progress in the following two areas: (1) There is consideration of a multi-task mode. Combined with the correlation between each section, the prediction performance of the water quality of the section is enhanced by sharing and learning from each other of the water quality information of multiple sections at the same time. (2) The proposed model applies a hybrid structure of deep learning and multi-task models, using deep convolutional neural networks (CNN), LSTM and multi-task learning (MTL) to achieve not only information sharing, but also the ability to be integrated with specific information from different sections. This new hybrid model can not only identify the nonlinear relationship between complex time-level inputs, but also can effectively obtain partial to comprehensive water quality information through the collaborative work between sections, further improving the prediction accuracy of the model.

The remainder of this essay is structured as follows: the second part is the pollution status control of the Lanzhou portion of the Yellow Rive r and the data description used by the MTL-CNN-LSTM model. The third part is a detailed introduction of the MTL-CNN-LSTM model discussed in this paper. The fourth part provides a thorough evaluation and discussion of the experimental results for the prediction of the multi-task prediction model on the four sections of the Lanzhou portion of the Yellow River. Finally, Section 5 draws conclusions.

## 2. Study Area and Materials

Lanzhou is the only city in the country where the Yellow River runs through the city. The main stream of the Yellow River is the largest river in Gansu Province, with a drainage area of 85,000 km$^2$. The Lanzhou section of the Yellow River starts from Liujiaxia Reservoir in Yongjing County in the west and ends at Wufo Temple in Jingtai County in the east, with a total length of 358 km. The Lanzhou urban river section is located in the Lanzhou Basin and presents an east–west trend. It passes through Xigu District, Anning District, Qilihe District, and Chengguan District, flows through southern Gaolan County and northern Yuzhong County, and exits at Wujinxia, with a total length of 152 km. With economic growth and an increase in the level of urban industrialization, the water pollution of the Lanzhou section of the Yellow River is becoming more and more serious, which not only affects the function of the Yellow River water ecosystem, but also seriously affects the environmental ecological balance of the Yellow River Basin. Lanzhou Environmental Monitoring Station monitors sections of the Yellow River all year round, and the section distribution in this study is consistent with that of the Environmental Monitoring Station. The specific section distribution is shown in Figure 1.

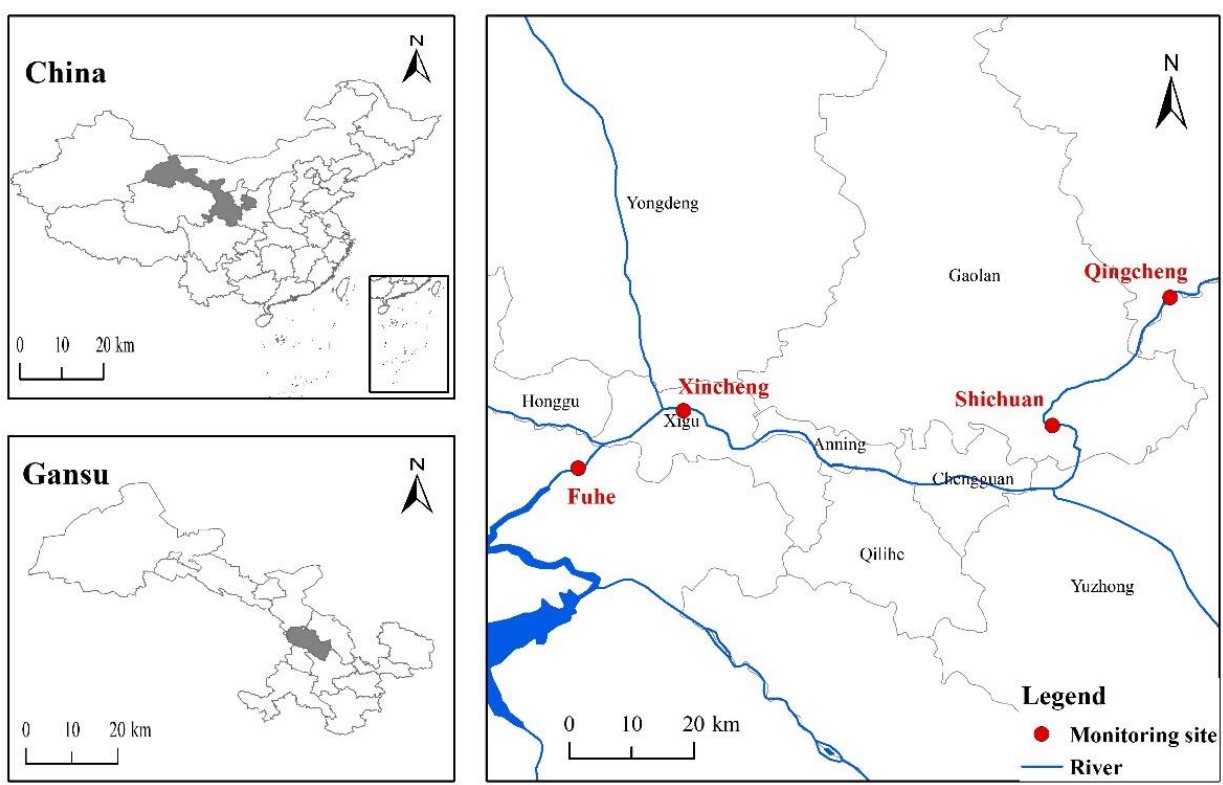

**Figure 1.** Cross-sectional distribution map of the Lanzhou portion of the Yellow River.

This study obtained the monitoring water quality data of the Lanzhou portion of the Yellow River from January 2018 to December 2020 (36 months) from Gansu Provincial Institute of Environmental Science and Design as the model data set for this study, including water pH, ammonia nitrogen concentration, potassium permanganate index, chemical oxygen demand (COD), total nitrogen, and total phosphorus 6 water quality indicators. The obtained water quality data are compared with the "Surface Water Environmental Quality Standard", and it is found that the chemical oxygen demand content of the Lanzhou section of the Yellow River has the greatest impact on its water quality category, so the chemical oxygen demand is the main research object for estimating the water quality of the Yellow River. A sample data set is shown in Table 1.

**Table 1.** Cross-section monitoring data of the Lanzhou section of the Yellow River.

| Station ID | Time | PH (PPM) | COD (PPM) | KMnO$_4$ (PPM) | NHN (PPM) | TP (PPM) | TN (PPM) |
|---|---|---|---|---|---|---|---|
| 2 | 28/3/2019 8:00:00 | 7.72 | 10.31 | 1.59 | 0.17 | 0.039 | 2.13 |
| 2 | 28/3/2019 12:00:00 | 7.71 | 10.38 | 1.48 | 0.16 | 0.046 | 2.02 |
| 2 | 28/3/2019 16:00:00 | 7.67 | 10.20 | 1.60 | 0.13 | 0.046 | 2.07 |
| 2 | 28/3/2019 20:00:00 | 7.67 | 10.21 | 1.60 | 0.15 | 0.037 | 2.00 |
| … | … | … | … | … | … | … | … |

When the model is trained on the cross-section, all the sections are simultaneously studied for multiple section tasks. In the experiment, the first 28 months of the data set monitored by the cross-section are used as the training set, 3 months are used as the validation set, and 3 months are used as the test set. The data from the previous 24 h are the input for the trained model, and it predicts the COD index in the following hour. Mean absolute error (MAE) and root mean square error (RMSE) are two evaluation criteria used

to assess the effectiveness of the proposed prediction model. The following definition of MAE, which reflects the model's prediction accuracy:

$$MAE = \frac{1}{M_n} \sum_{t=1}^{Mn} |y_t^n - \hat{y}_t^n| \tag{1}$$

where $M_n$ is the number of test samples, $y_t^n$ and $\hat{y}_t^n$ are the predicted value of the model and the true value of sample $t$ at section $n$, respectively. The smaller the MAE, the smaller the mean of the absolute error between the predicted and observed values, indicating a better model. RMSE assesses the stability of the model and is characterized as:

$$RMSE = \sqrt{\frac{1}{Mn} \sum_{t=1}^{Mn} (y_t^n - \hat{y}_t^n)^2} \tag{2}$$

where $y_t^n$ and $\hat{y}_t^n$ are the true value of the model and the predicted value, respectively. A smaller RMSE value means a better model.

### 3. Methodology Research and Design

Multi-task learning (MTL) is a type of migration algorithm [20]. Multiple related tasks are learned in parallel at the same time, through the underlying shared representation and complementary learned domain-related information, the generalization performance of the model is improved. Moreover, during single-task learning, the propagation of gradients is easy to fall into local minima; but in MTL, the local minima of different tasks are in different positions, and they can help escape from local minima through interaction; therefore, multi-task learning also has a better feature learning capability. In recent years, it has drawn an increasing amount of attention. This study based on the COD concentration series of the Yellow River monitored by each section as the test data, the Pearson correlation coefficient was used to analyze the relationship between multiple sections. Table 2 presents the outcomes. The correlation of water quality among four sections of the Lanzhou portion of the Yellow River provides the possibility of multi-task learning for water quality prediction of multiple sections.

**Table 2.** Section correlation analysis chart.

| Station ID | 1 | 2 | 3 | 4 |
|---|---|---|---|---|
| 1 | 1.00 | 0.53 | 0.56 | 0.83 |
| 2 | 0.53 | 1.00 | 0.66 | 0.53 |
| 3 | 0.56 | 0.66 | 1.00 | 0.72 |
| 4 | 0.83 | 0.53 | 0.72 | 1.00 |

Deep learning is currently more and more popular in the field of artificial intelligence and is widely used in various fields, including speech recognition, natural language processing, computer vision, autonomous driving and prediction work, and its prediction performance is better than other previous machine learning models [24,25]. Due to the complexity of water pollution, the fluidity of water and the variability of water quality, the water quality prediction problem lacks certain accuracy and generalization. Deep learning techniques can better obtain these features through hidden layers with machine learning models and large amounts of training data. The convolutional neural network, one of the deep learning algorithms, has the properties of local connection, weight sharing, and pooling downsampling, which lower the complexity of the network model. It has a good picture recognition performance and can extract and categorize the feature information from the image data with ease. Inspired by it, scholars have explored the application of CNN in time series prediction, and found that the feature extraction function of CNN through convolution operation can also be applied to time series data analysis, and the noise tolerance in time series prediction is very good [26]. The prediction of time series uses a one-dimensional CNN. The convolution kernel can be regarded as a window, which

translates the time series data, extracts local sequence segments and multiplies them with weights, and continuously outputs the calculated sequence features. Then, it can perform pooling downsampling to further filter the noise information that is not useful for prediction in the data and extract more valuable information that is more useful for the final water quality prediction, so that the prediction performance is optimized.

The CNN model alone can perform convolution operations on each period of time sequence to extract local features in water quality information, but CNN is not sensitive to the time sequence of water quality and cannot complete the overall prediction task well alone. Water quality information is affected by many factors and has a nonlinear trend, so it is difficult to predict water quality. As a variant of Recurrent Neural Network (RNN) [27], LSTM combines the gating function and hidden state to solve the problem of RNN gradient disappearance, and each neural unit of LSTM consists of input gate, forget gate, output gate, memory cell state and hidden unit. It consists of three gates and two fundamental units. Through this structure, it can be determined which information may be forgotten and which information may be maintained. It is advantageous for handling time series information, such as water quality prediction problems [28], air quality prediction problems [29] and the stock market forecasting problem [30], but utilizing the LSTM model alone will introduce noisy data that have no relationship with water quality predictions, resulting in poor predictions. Therefore, this paper integrates CNN and LSTM to build an end-to-end deep learning network, which makes full use of the feature information extraction ability of CNN and the sensitivity of LSTM to time series data to enhance the effect of water quality prediction.

Therefore, this chapter proposes a multi-task water quality prediction model based on deep learning (MTL-CNN-LSTM), which can obtain multi-section water quality characteristics, realizes the induction and migration between sections, and achieves the effect of complementing each other. The model includes shared coding layer based on CNN, specific time series feature layer based on LSTM and loss parameter optimization. First, each section uses the deep neural network model as the bottom layer to learn their common features, so as to realize the sharing of feature information between multiple sections; then, the respective features of different sections are extracted through their specific task modules to ensure the independence of tasks. The advantage of doing this is to share the water quality information of each section while maintaining the independence of each section to obtain more in-depth water quality characteristics. Secondly, considering that the input water quality data have both local dependencies and long-term dependencies of time series between them, the order of CNN-LSTM is used to design, not only will not disrupt the long-term dependency information of time series, but can also better mine temporal features from local to full, thereby improving the overall performance of the model. The model structure is shown in Figure 2. The fundamental principles and experimental details of each part are specified as follows:

### 3.1. Shared Coding Layer Based on CNN

This module realizes the feature sharing and joint learning of the water quality data of the section. The hidden layers of CNN in this model include convolutional layers and pooling layers. The water quality data of various sections are input into the CNN for training simultaneously, in which the main features of the water quality data are extracted and filtered by convolutional and pooling layers, which not only achieves the sharing of parameters among various sections, but also reduces the possibility of overfitting among the respective sections.

In order to obtain the feature sequence representation among multiple section data, the entire one-dimensional convolution part can be regarded as a special data preprocessing structure [31], where the water quality information is convolved and refined into the input of the LSTM part that is more sensitive to time series information. The convolution process is shown in Figure 3 below.

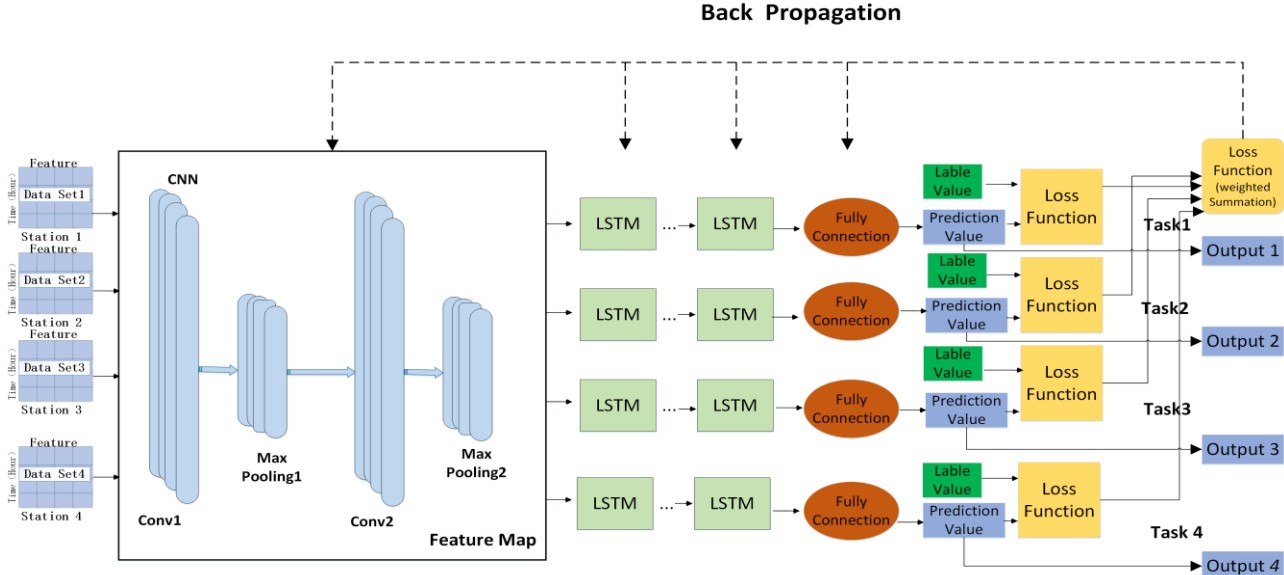

**Figure 2.** MTL-CNN-LSTM prediction model structure.

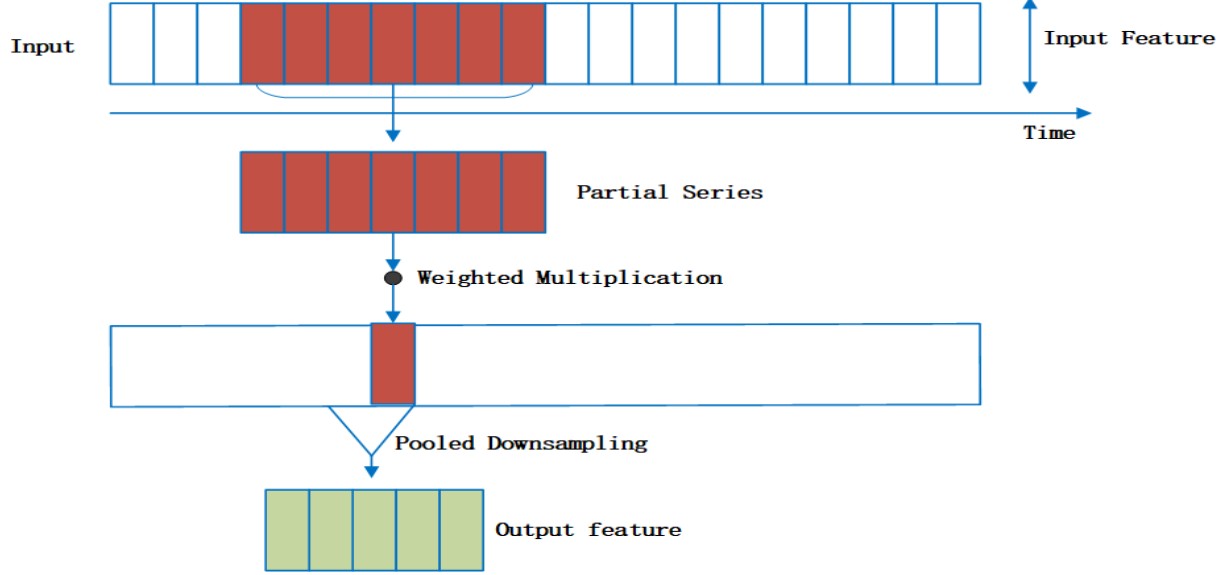

**Figure 3.** The calculation process of the convolution layer.

### 3.2. Specific Time Series Feature Layer Based on LSTM

The main purpose of this module is to learn the long-time dependent features between the water quality of each section. Since the model is an end-to-end structure, the shared features are extracted and entered into the respective LSTM model for multiple training to learn the time series feature information among the water quality of each section, so as to retain the unique features of the data of each site. Finally, the output vector of water quality information of each section obtained through multiple training is input to the fully connected layer to obtain the final prediction value.

The network unit of LSTM is shown in Figure 4. In model training, the two-dimensional vector $X(t)$ with time series characteristics and the state $H(t-1)$ at the previous moment are input into the cyclic unit structure for training. The activation function (sigmoid func-

tion) σ can obtain the gated signals of three gates, $F(t)$, $O(t)$, and $I(t)$, and candidate state $\widetilde{C}_t$. The calculation method for the three gates is:

$$F(t) = \sigma\left(W_f.\ [H_{t-1}, X_t] + b_f\right) \tag{3}$$

$$O(t) = \sigma(W_O.\ [H_{t-1}, X_t] + b_O) \tag{4}$$

$$I(t) = \sigma(W_i.\ [H_{t-1}, X_t] + b_i) \tag{5}$$

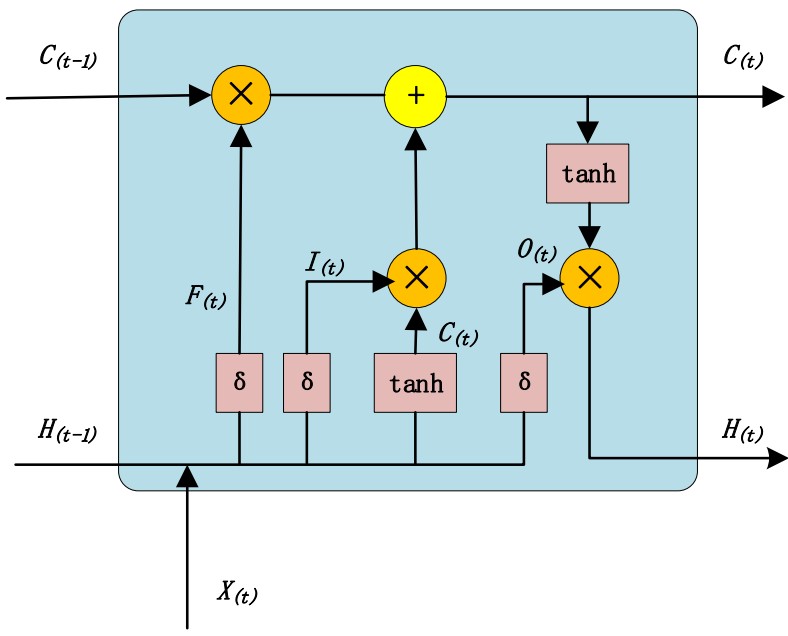

**Figure 4.** LSTM unit architecture.

Among them, $W_f$, $W_O$, and $W_i$ are the input weights of the matrix, $b_f$, $b_o$, and $b_i$ are the bias weights, $t$ represents the current time state, and $X$ represents the input information. $F(t)$ controls how much information is to be forgotten in the internal state $C_{t-1}$ at the previous moment. The closer $F(t)$ is to 0, the more information is forgotten, and the closer to 1, the more information is memorized. Therefore, $F(t)$ decides that the LSTM model will selectively forget some of the past COD data and other factor information. In the same way, $I(t)$ controls how much COD data and other factor information need to be saved in the candidate state $\widetilde{C}_t$ at the current moment. After obtaining the gate control signal, combine the forget gate $F(t)$ and the input gate $I(t)$ to update the memory cell state $C$:

$$C(t) = F_t \times C_{t-1} + I_t \times \widetilde{C}_t \tag{6}$$

$$\widetilde{C}_t = \tanh(W_c.\ [H_{t-1}, X_t] + b_c) \tag{7}$$

Combining the output gate $O(t)$ with the updated cell state $C$ at this moment, the information of the updated internal state can be transferred to the external state $H(t)$. LSTM will continue to run in a loop according to the basic unit until the end of the training:

$$H_t = O_t \otimes \tanh(C_t) \tag{8}$$

The feature vector $X_P$ extracted by CNN is input into the LSTM, the feature timing information is extracted twice through the LSTM, and finally the COD prediction result $\widetilde{y}_i^t$ of a certain section at time t is obtained after decoding through the fully connected layer:

$$L_t^u = g\left(w^T X_P + d^T L_t\right) \tag{9}$$

$$\widetilde{y}_i^t = f\left(\xi^T L_t\right) \tag{10}$$

Among them, $L^u$ is the output of the LSTM, $u$ is the weight matrix of the input layer to the convolutional layer, $w$ is the weight matrix from the feature $X_p$ to the LSTM layer, $d$ is the weight matrix of information transmission between neurons in the LSTM, $\xi$ is the weight matrix of LSTM to the fully connected layer, and $i$ represents the prediction task $i \in [1, m]$ of a section.

### 3.3. Multi-Loss Joint Optimization

In the loss joint optimization design of this paper, since multiple sections are trained to predict the model simultaneously, and the loss function results of multiple cross-section tasks are used for joint training, more cross-section correlations can be obtained by using the label data of different cross-section tasks, thereby further improving the predictive ability of the model. This paper establishes $N$ prediction tasks with the same task weight, $D_n$ is the training set of task $n$, and the number of training samples is $M_n$. The training set can be expressed as:

$$D_n = \left\{ \left( x^{(n,m)}, y^{(n,m)} \right) \right\}_{m=1}^{Mn} \tag{11}$$

Among them, $x^{(n,m)}$ and $y^{(n,m)}$ represent the $m$th sample and its label in the $n$th task. In this model, since multiple prediction tasks are carried out simultaneously, there are multiple output values and actual values, so there are multiple loss functions. We use the weighted combination of the loss functions of multiple tasks as the overall optimization loss function of the prediction model, so as to use the backpropagation algorithm to jointly optimize the model, and finally obtain the output values of multiple prediction tasks through multiple iterative training and error correction. The prediction of water quality in this paper is a numerical regression problem. The mean square error (MSE) is a commonly used loss function in the gradient descent calculation of the regression prediction model. The loss function $\zeta_n$ of the nth section is expressed as:

$$\zeta_n = \frac{1}{Mn} \sum_{t=1}^{Mn} (y_t^n - \hat{y}_t^n)^2 \tag{12}$$

$y_t^n$ is the predicted value of task $n$ at time t, $\hat{y}_t^n = f_n(x_t^n, \theta)$, $\theta$ represents all parameter sets, including shared layer and specific layer. The importance of each task in the model is the same. The linear weighted sum of all task loss functions is used as the joint objective function of multi-task learning. The joint optimization loss function $\zeta$ of the overall model is:

$$\zeta = \sum_{m=1}^{M} \eta_m \zeta_m \tag{13}$$

$\eta_m$ is the weight of the $m$th task, Since this model needs to take into account all tasks, the risk of overfitting is avoided to a certain extent.

### 3.4. Training Process and Algorithm of the Model

There are five main steps in the experimental study on the model. The key steps are described below.

Step 1: **Determine the number of tasks for the model.** Use the Pearson correlation coefficient to test the correlation between sections, and then predict M sections at the same time.

Step 2: **Input**. Input the data set $D$ of $m$ sections into the model, the input feature vector of each section is $D_{T-1}, \ldots, D_{T-R}$, and perform COD index prediction for m sections at the same time.

Step 3: **Extracting features.** Input vectors of m sections are extracted by CNN for local feature information on the input matrix, multiple sections in the shared layer achieve parameter sharing.

Step 4: **Decoding.** Input the extracted feature information into their corresponding LSTM for multiple iteration training and extract the time series features of *m* sections. Finally, the output of the LSTM is connected to the fully connected layer to obtain the predicted value of the model.

Step 5: **Model optimization.** The loss functions of the m tasks are weighted jointly, and the overall optimization of the model is performed by back propagation.

Algorithm 1 describes the equivalent algorithm for the MTL-CNN-LSTM model training procedure mentioned above.

---

**Algorithm 1:** The MTL-CNN-LSTM prediction method

---

**Collect** water quality data between sections and establish a water quality database
**Use** the Pearson correlation coefficient to test the correlation between sections
**Build** the MTL-CNN-LSTM prediction model
**Input:** training data set $D_m (1 \leq m \leq M)$ of $M$ section tasks;
upper limit of learning epochs $S$; learning rate $\alpha$
1: initialize all adjustable parameters $\theta_0$ randomly
2: Enter training
3: **for** $S = 1$ to $S$ do
4:   **for** $m = 1$ to $M$ do
5:      $X^l = CNN_{shared}(D_m)$  //shared layer
6:      $y_t^m = LSTM\left(X^l\right)$       //task-specific layer
7:      $\zeta_m = \frac{1}{K_m} \sum_{t=1}^{K_m} (y_t^m - \hat{y}_t^m)^2$  //calculate the loss $\zeta_m$ of task $m$
8:   **end for**              // Terminate "for"
9:   $\zeta = \sum_{m=1}^{M} \sum_{k=1}^{K_m} \eta_m \zeta_m$       // calculate the overall loss $\zeta$
10:   $\theta_0 \leftarrow \theta_{k-1} - \alpha \cdot \nabla \theta \zeta(\theta)$
11:   **if** $\zeta$ stop reducing for more than 100 times **then**
12:      **break**
13:   **end if**               // Terminate "if"
14: **end for**              // Terminate "for"

---

## 4. Results and Discussion

### 4.1. Experimental Environment and Parameter Configuration

The hyperparameter settings of the MTL-CNN-LSTM model affect its prediction performance of the Yellow River water quality to a certain extent. After repeated experiments, relatively good hyperparameters (Table 3) and activation functions were determined.

**Table 3.** The MTL-CNN-LSTM model hyperparameter settings.

| Model Component | Kernel Size | Number of Convolution Kernels | Number of Parameters |
|---|---|---|---|
| Convolutional layer 1 | $7 \times 1$ | 6 | 42 |
| Max Pooling layer 1 | $2 \times 1$ | | 0 |
| Convolutional layer 2 | $7 \times 1$ | 14 | 98 |
| Max Pooling layer 2 | $2 \times 1$ | | 0 |
| Convolutional layer 3 | $7 \times 1$ | 28 | 420 |
| Max Pooling layer 3 | $2 \times 1$ | | 0 |
| LSTM | | 50 | 15,800 |
| Dense layer | | 1 | 51 |

The implementation of each experiment in this paper is based on Keras2.3.1 of Tensorflow2.0.0, a deep learning framework developed by Google team. During the experiment, the training set, validation set, and test set were set according to the ratio of 8:1:1. The MTL-CNN-LSTM model used random initialization parameters, and Adam was used as the optimization algorithm. The learning rate $\alpha$ is 0.05, the weight ratio of each prediction section is the same, and batch_size is 72.

*4.2. Comparative Analysis of Model Performance*

Compare the MTL-CNN-LSTM with the deep learning single-task model with outstanding performance in time series.

The description of the comparative prediction model is as follows:

MTL-CNN-LSTM: Use a multi-task water quality prediction model to make simultaneous predictions based on four sections of the Lanzhou portion of the Yellow River. It has a multiple-input multiple-output structure. LSTM: It has a single-input single-output structure and only considers time series features.

CNN-LSTM: Predicting water quality of a single section using a hybrid model, with a single-input single-output structure.

LSTM: Considering global time series features, it has a single-input single-output structure.

CNN: Convolution operation is performed on each time series using one-dimensional convolution, which has a structure of single input and single output.

The following is a comparative analysis of the performance of different models at t + 1.

As shown in Table 4, the predicted values of the CNN model and the LSTM model have a large deviation from the actual value as a whole. Under the same data set, the prediction performance of a single model of CNN and LSTM is worse than that of the fusion model CNN-LSTM. The ability of a convolutional neural network to automatically extract data features can better capture the short-term local feature information of water quality time series data; LSTM is more sensitive to time series data and can better capture the long-term dependence information of water quality time series data. Thus, the two fusion prediction models outperformed the single model in the prediction of water quality indicators.

**Table 4.** Prediction performance evaluation index of different models at t + 1.

| Model | Station | | | | | | | | | |
|---|---|---|---|---|---|---|---|---|---|---|
| | 1 | | 2 | | 3 | | 4 | | Mean | |
| | MAE | RMSE | MAE | RMSE | MAE | RMSE | MAE | RMSE | MAE | RMSE |
| MTL-CNN-LSTM | 0.250 | 0.312 | 0.294 | 0.380 | 0.224 | 0.288 | 0.279 | 0.304 | 0.262 | 0.321 |
| CNN-LSTM | 0.342 | 0.421 | 0.417 | 0.479 | 0.235 | 0.296 | 0.214 | 0.323 | 0.302 | 0.380 |
| LSTM | 0.379 | 0.452 | 0.446 | 0.517 | 0.386 | 0.438 | 0.494 | 0.509 | 0.426 | 0.479 |
| CNN | 0.394 | 0.472 | 0.584 | 0.664 | 0.448 | 0.491 | 0.542 | 0.567 | 0.492 | 0.549 |

The prediction effect of the MTL-CNN-LSTM discussed in this paper is better than the prediction effect of the single-task model on the four sections. This is because the MTL-CNN-LSTM can share water quality features between multiple sections, learn the data contained in multiple sections in the shared layer, can dig out more and more effective time series and spatial characteristics, so the model has a better prediction effect. Figure 5 shows the fitting effect of the MTL-CNN-LSTM model on the four sections.

In Table 4 mean represents the average of the prediction results of each model on the four sections, and the average prediction results of each model in turn are used as the reference benchmark to calculate the improvement rate of each model on MAE and RMSE indexes from Table 4 and the results are shown in Figure 6. The MAE and RMSE of LSTM are lower than CNN by 13.4% and 12.6%, respectively, and this shows that for the prediction of time series, adding time features to the prediction model can help improve the prediction accuracy of the model. The MAE and RMSE of CNN-LSTM are lower than LSTM by 29.1% and 20.7%, respectively, and the overall performance is higher than that of the LSTM model. It shows that the convolutional neural network can automatically extract the local features of the data, can filter noise, and capture the impact of water quality features on the water quality indicators of the section, which can effectively improve the prediction performance of the model and obtain more accurate results. The MTL-CNN-LSTM proposed in this paper has the best performance among the four models. The MAE and RMSE of the MTL-CNN-LSTM are lower than CNN-LSTM by 13.2% and 15.5%, respectively, and it shows that simultaneous prediction of water quality monitoring sections

with strong spatial correlations can significantly raise the prediction performance of the model on water quality.

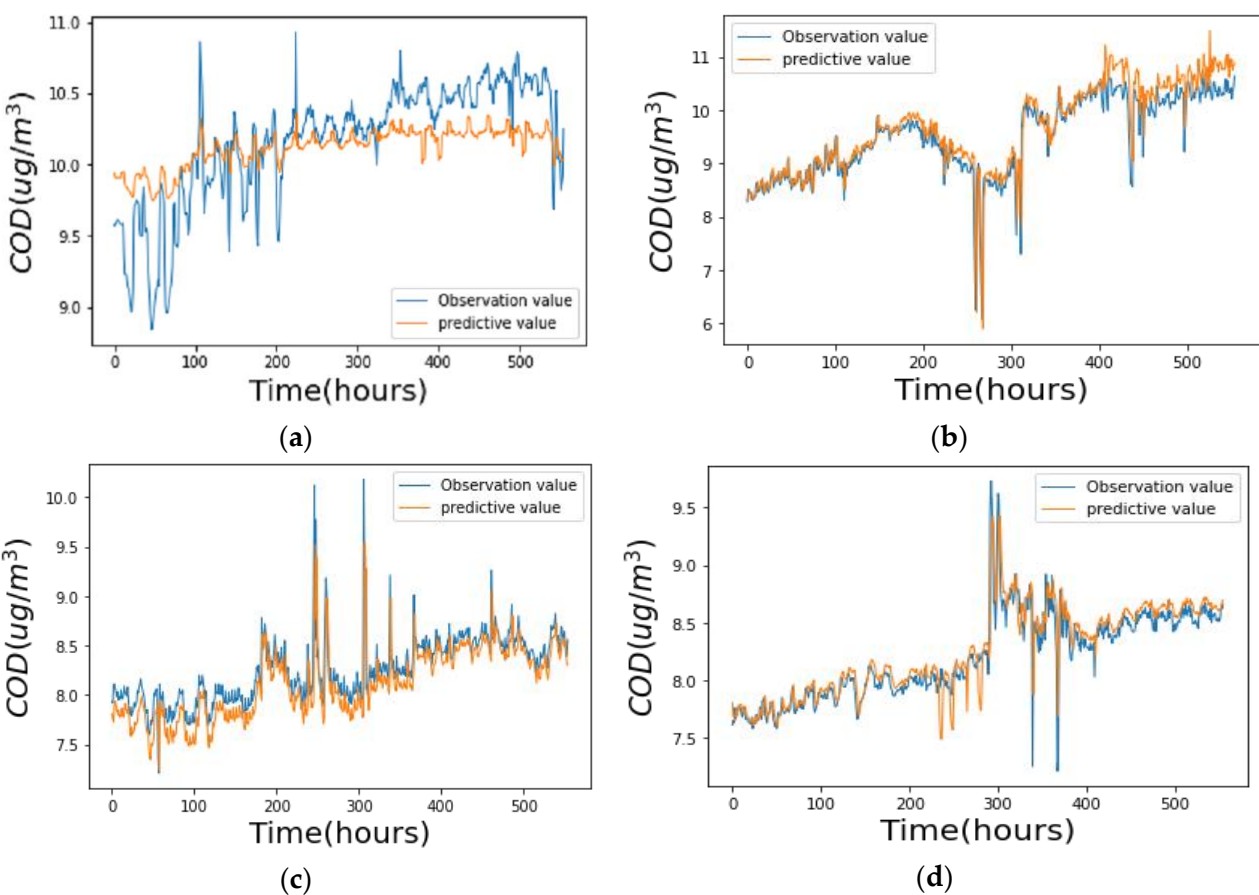

**Figure 5.** Fitting comparison of the MTL-CNN-LSTM model under each section. (**a**) Model fit trend comparison on Section 1. (**b**) Model fit trend comparison on Section 2. (**c**) Model fit trend comparison on Section 3. (**d**) Model fit trend comparison on Section 4.

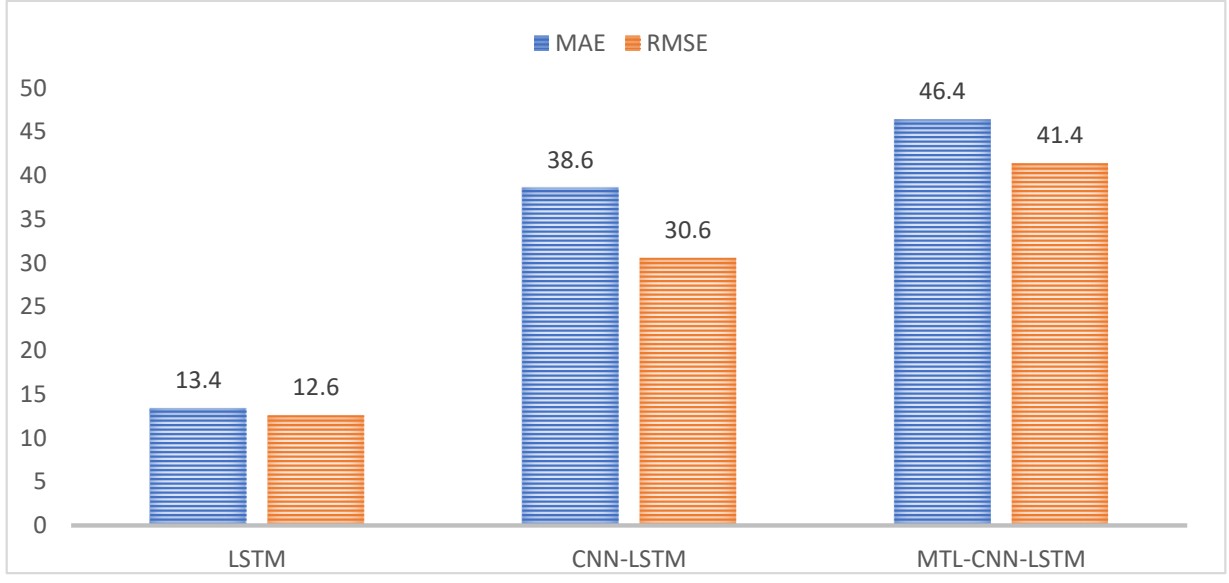

**Figure 6.** Schematic diagram of model performance improvement efficiency comparison.

## 5. Conclusions

In this paper, a water quality prediction model based on multi-task deep learning (MTL-CNN-LSTM) is proposed, taking four sections of the Lanzhou portion of the Yellow River as the research objects, and applying it to the 36-month monitoring data set. The results show that:

(1) Due to the fluidity of water, there is a correlation between sections within a certain range. In the process of model training, a multi-task learning method is added to retain specific information between sections and also share similar information, which can better facilitate the deep mining of the Yellow River water quality index and improve the accuracy of the model.

(2) Water quality prediction is a typical time series problem. The MTL-CNN-LSTM is used to predict the COD index of the Yellow River; first, the convolutional layer is used to better extract the local characteristics of water quality between sections of the Yellow River, and then the LSTM model is used to obtain the correlation characteristics between the long-term dependencies of the water quality of multiple sections, and the water quality information between the sections is obtained from the local to the whole to improve the accuracy of the prediction model.

The MTL-CNN-LSTM can simultaneously predict multiple sections of the Lanzhou portion of the Yellow River, and the prediction accuracy is better than the single-section prediction of a single model and can effectively fit the complex changes of the COD sequence in the Yellow River water. However, due to the small data set in this study, it is limited to the monitoring data of the Lanzhou portion of the Yellow River, which may have an impact on model training. The next research will consider more river water quality information and conduct classification and prediction research on specific water quality indicators in specific regions. Ultimately, our goal is to develop this model into an application prediction model, which can provide new methods for basin water quality management and a reliable basis for river prevention and control, and effectively improve the efficiency of water pollution monitoring.

**Author Contributions:** F.W. collected the data. Q.Z. is the designer of the method to guide the experimental design and data analysis. X.W. is the executor of this experimental study, completing data analysis and writing the manuscript. Y.Q. is mainly responsible for guiding the manuscript revision. All authors have read and agreed to the published version of the manuscript.

**Funding:** This research was supported by Gansu Provincial Science and Technology Program (Project No. 22JR5RA176) and Northwest Normal University 2021 Young Faculty Research Capacity Enhancement Program (Project No. NWNU-LKQN2021-21).

**Data Availability Statement:** The data presented in this study are available on request from the corresponding author. The data are not publicly available due to privacy issues.

**Conflicts of Interest:** The authors declare that they have no known competing financial interests or personal relationships that could have appeared to influence the work reported in this paper.

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
