# Peer review of "A Water Quality Prediction Model Based on Multi-Task Deep Learning: A Case Study of the Yellow River, China"

_water, doi:10.3390/w14213408_

Round 1
Reviewer 1 Report
The manuscript must be revised by an English as a second language editorial service.
The Abstract and Introduction are poorly written. The Study area and materials section should be revised to remove results from this section. The Research methodology and models section should be moved to Supplementary Information. Sections 3 and 4 are better written in English than other sections, i.e., Lines 272–290. It makes me wonder if different authors wrote separate sections of the manuscript or if parts of the manuscript were published previously.
Abstract:
Lines 10–11: Sentence beginning “However,…” makes no sense in English.
Line 14: What does MTL-CNN-LSTM stand for? You finally defined “MTL” in line 217 after using the abbreviation several times. This is not acceptable.
Introduction:
Delete the use of the word “Nowadays” in lines 28 and 32.
Line 36: Avoid the use of one-sentence paragraphs.
Line 39: Incomplete sentence.
Line 42: Capitalize “DO.”
Line 48: GM? Define all acronyms at first use.
Line 62: RBF? ….and many more undefined acronyms throughout the manuscript.
Study area and materials:
In Figure 1, it is difficult to follow the path of the Yellow River in the central part of the figure. Revise this figure.
Tables 1 and 2 are Results (see line 196) and do not belong in the study area and materials section. They should be moved to section 3.
Research methodology and models (this section could be moved to supplementary information):
Lines 227–265: This paragraph is too long and should be divided.
Can the authors assure this reviewer that Figures 2, 3, and 4 are original figures to this manuscript, prepared by the authors, and never published before elsewhere?
In Algorithm 1, the purposes/description of the “end for” and “end if” commands must be explained.
Lines 409–410: What is the purpose in this manuscript for this statement?
Lines 413–414: If this refers to software, the company must be acknowledged.
Author Response
Dear reviewer:
We really appreciate you for your carefulness and conscientiousness. Your suggestions are really valuable and helpful for revising and improving our paper. According to your suggestions, we have made the following revisions on this manuscript:
Point 1: The manuscript must be revised by an English as a second language editorial service.
Response 1: We appologize for the language problems in the original manuscript. Considering the Reviewer’s suggestion, we will take great effort to modify the full text to make it more professional.Revised manuscript has been uploades, and I hope it can meet with requirement.
Point 2: Lines 10–11: Sentence beginning “However,…” makes no sense in English.
Response 2: As Reviewer suggested that this sentence make no sense and it was rectified at Line 9-12.
Point 3: Line 14: What does MTL-CNN-LSTM stand for? You finally defined “MTL” in line 217 after using the abbreviation several times. This is not acceptable.
Response 3: We are grateful for the suggestion. As suggested by the reviewer, we have reworked the Abstract section and explained when the MTL was first appeared in the main text(see Line 98).
Point 4: Delete the use of the word “Nowadays” in lines 28 and 32.
Response 4: We have made correction according to the Reviewer’s comments and it was rectified at Line 28-34.
Point 5: Line 36: Avoid the use of one-sentence paragraphs.
Response 5: Thank you for pointing out this problem in manuscript. We have rectified at Line 35-37.
Point 6: Line 39: Incomplete sentence.
Response 6: Thank you for pointing out this problem in manuscript. We have rectified at Line 37-38.
Point 7: Line 62: RBF? ….and many more undefined acronyms throughout the manuscript.
Response 7: Thank you for pointing out this problem in manuscript. We carefully revised undefined abbreviations and explained them when they first appeared in the text. We have rectified at Line 42: FRHI, Line 52: GWLF, Line 50: DIN, Line 68: ARIMA and GAWNN.
Point 8: In Figure 1, it is difficult to follow the path of the Yellow River in the central part of the figure. Revise this figure.
Response 8: According with your advice, we have prepared through RS-GIS as shown in Figure 1.
Point 9: Tables 1 and 2 are Results (see line 196) and do not belong in the study area and materials section. They should be moved to section 3.
Response 9: We are grateful for the suggestion. As suggested by the reviewer, we have moved Table 2 to section3 (see Line 197). However, Table 2 shows the research data, and we think it should be more suitable for the study area and materials section.
Point 10: Research methodology and models (this section could be moved to supplementary information).
Response 10: We are grateful for the suggestion. The main highlight of this article is water quality prediction through the use of deep learning, as expressed in the Line 130-136,each section of the article is interconnected.So we believe that Methodology Research and Design is more appropriate for the third part.
Point 11: Lines 227–265: This paragraph is too long and should be divided.
Response 11: Thank you for pointing out this problem in manuscript. We have rectified at Line 220-241, Line 242-259.
Point 12: Can the authors assure this reviewer that Figures 2, 3, and 4 are original figures to this manuscript, prepared by the authors, and never published before elsewhere?
Response 12: Yes, we make sure that Figures 2, 3, and 4 are original figures to this manuscript, prepared by the authors, and never published before elsewhere.
Point 13: In Algorithm 1, the purposes/description of the “end for” and “end if” commands must be explained.
Response 13: Thank you for pointing out this problem in manuscript. In Algorithm 1, “end if” and “end for” represent the termination of the “if” and “for”.We have made correction according to the Reviewer’s comments(see Line 367).
Point 14: Lines 409–410: What is the purpose in this manuscript for this statement?
Response 14: We are very sorry for our incorrect writing this statement, we have deleted it.
Point 15: Lines 413–414: If this refers to software, the company must be acknowledged.
Response 15: Thank you so much for your careful check. TensorFlow is one of the deep learning frameworks developed by the Google team to train and run deep neural networks. We have rectified at Line 375-376.
We look forward to your information about my revised paper.Once again, thank you very much for your comments and suggestions.
Yours sincerely,
Ying Qi

Reviewer 2 Report
Dear authors,
As per my view, manuscript having enough novelties and could be considered for publication after major revision requested below:
Introduction written very lengthy and need to be focus on material related to the theme of the study (kindly revise it with suggested references; Gupta et al., https://doi.org/10.1007/978-981-13-8181-2_8; Pandey et al., Springer, Cham. https://doi.org/10.1007/978-3-030-51427-3_23)
Line no 40-42: need to be supported with (A novel approach for river health assessment of Chambal using fuzzy modeling, India; https://doi.org/10.5004/dwt.2017.0144)
Figure 1 need to be prepared through RS-GIS
Implementation of this study to regional river or global river is missing
Author Response
Dear reviewer:
We really appreciate you for your carefulness and conscientiousness. Your suggestions are really valuable and helpful for revising and improving our paper. According to your suggestions, we have made the following revisions on this manuscript:
Point 1: Introduction written very lengthy and need to be focus on material related to the theme of the study
Response 1: We are grateful for the suggestion. According with your advice, we have referred and added relevant literature. Please see Line 46-49 and Line 58-62.
Point 2: Line no 40-42: need to be supported with (A novel approach for river health assessment of Chambal using fuzzy modeling, India; https://doi.org/10.5004/dwt.2017.0144)
Response 2: As suggested by the reviewer, we have added this reference in the introduction part and hope that it is now clearer.Please see Line41-44.
Point 3: Figure 1 need to be prepared through RS-GIS
Response 3: According with your advice, we have prepared through RS-GIS as shown in Figure 1.
Point 4: Implementation of this study to regional river or global river is missing.
Response 4: We are grateful for the suggestion. The highlight of this article is to improve the effect of water quality prediction models by using deep multi-task learning. However, due to the dataset, we can only experiment in Lanzhou portion of the Yellow River at present,and we are aware of this problem and point out in the discussion section. We will expand the dataset and apply this study to a larger river in subsequent experiments.
We look forward to your information about my revised paper.Once again, thank you very much for your comments and suggestions.
Yours sincerely,
Ying Qi

Reviewer 3 Report
This paper is interesting and quite comprehensive and adequate. This paper deals with a a multi-task water quality prediction model based on deep learning (MTL-CNN-LSTM), using the chemical oxygen demand (COD) month data of the water environment of the Lanzhou section of the Yellow River as research object.
Strengths - presents an ambitious goal, a water quality prediction model based on deep learning (MTL-CNN-LSTM), retaining spatial heterogeneity. Authors state a reduction of mean absolute error and root mean square error of the predicted value of the model (13.2% and 15.5%) with better time stability and generalization performance.
Weakness – very small data set, particularly limited in time (36 months) and space (just 4 sections).
Abstract – the information given is adequate and concise.
1.Introduction – The aim of paper is adequately exploited. Adequate references.
Line 40-42 – correct “Dissolved oxygen (DO) levels in a riverine environment, in Calgary, Canada were predicted using a fuzzy linear regression method, this method improves the prediction of 41 low Do DO and thus…..”
2. Materials –adequate
Fig 1 – should be improved. It should also have a scale and the latitude/longitude data.
Table 1 – insert units for COD, KMnO4, NHN, TP and TN.
3. Results - Adequate and well documented by figures.
Line 282 - correct sentence
4. Discussion – Adequate and a thorough comparative analysis of the multi-task water quality prediction model based on deep learning (MTL-CNN-LSTM) with the deep learning single-task model is quite useful.
5. Conclusions – Adequate.
Therefore in my opinion this paper should be accepted with minor revision.
Author Response
Dear reviewer:
We really appreciate you for your carefulness and conscientiousness. Your suggestions are really valuable and helpful for revising and improving our paper. According to your suggestions, we have made the following revisions on this manuscript:
Point 1: Weakness – very small data set, particularly limited in time (36 months) and space (just 4 sections).
Response 1: We are grateful for the suggestion. We are aware of the problem of the small data set and have presented it in the final discussion section, these are all the data currently available and we will add more data to optimize the model in the next step.
Point 2: Fig 1 – should be improved. It should also have a scale and the latitude/longitude data.
Response 2: According with your advice, we have prepared through RS-GIS as shown in Figure 1.
Point 3: Table 1 – insert units for COD, KMnO4, NHN, TP and TN.
Response 3: We are grateful for the suggestion. As suggested by the reviewer, we have inserted units for COD, KMnO4, NHN, TP and TN(see Table 1).
Point 4: Line 282 - correct sentence
Response 4: Thank you so much for your careful check. We have rectified at Line 249-250.
We look forward to your information about my revised paper.Once again, thank you very much for your comments and suggestions.
Yours sincerely,
Ying Qi

Reviewer 4 Report
Dear Authors,
excellent work, however Introduction part is to be added by international additional authors to avoid, that just part of the world is represented. I suggest adding Moore et al 2010 Modelling the Effects of Climate change on... as well as Pehme et al 2019 Urban hydrology research fundamentals, and Tamm et al 2008 Contributions of DOC from surface and groundflow...
Also The Objective should be more clearly specified
otherwise Minor changes
Author Response
Dear reviewer:
We really appreciate you for your carefulness and conscientiousness. Your suggestions are really valuable and helpful for revising and improving our paper. According to your suggestions, we have made the following revisions on this manuscript:
Point 1: I suggest adding Moore et al 2010 Modelling the Effects of Climate change on... as well as Pehme et al 2019 Urban hydrology research fundamentals, and Tamm et al 2008 Contributions of DOC from surface and groundflow...
Response 1: As suggested by the reviewer, we have added this reference in the introduction part and hope that it is now clearer.Please see Line49-52,reference 1 and Line 44-46.
Point 2: The Objective should be more clearly specified
Response 2: We are grateful for the suggestion. The highlight of this article is to improve the effect of water quality prediction models by using deep multi-task learning. As suggested by the reviewer, we have rectified at Line 126-129.
Point 3: otherwise Minor changes
Response 3: We are grateful for the suggestion. According with your advice we have modified the text to make it more professional.
We look forward to your information about my revised paper.Once again, thank you very much for your comments and suggestions.
Yours sincerely,
Ying Qi

Round 2
Reviewer 1 Report
The authors have extensively revised this manuscript. It is now acceptable for publication.